# Uncovering the effects of model initialization on deep model generalization: A study with adult and pediatric chest X-ray images

**Sivaramakrishnan Rajaraman**⬤**, Ghada Zamzmi**⬤**, Feng Yang**⬤**, Zhaohui Liang, Zhiyun Xue**⬤**, Sameer Antani**⬤*

Computational Health Research Branch, National Library of Medicine, National Institutes of Health, Bethesda, Maryland, United States of America

* sameer.antani@nih.gov

**Data Availability Statement:** All data used for this study are in the manuscript and it's supporting information files.

## Abstract

Model initialization techniques are vital for improving the performance and reliability of deep learning models in medical computer vision applications. While much literature exists on non-medical images, the impacts on medical images, particularly chest X-rays (CXRs) are less understood. Addressing this gap, our study explores three deep model initialization techniques: Cold-start, Warm-start, and Shrink and Perturb start, focusing on adult and pediatric populations. We specifically focus on scenarios with periodically arriving data for training, thereby embracing the real-world scenarios of ongoing data influx and the need for model updates. We evaluate these models for generalizability against external adult and pediatric CXR datasets. We also propose novel ensemble methods: F-score-weighted Sequential Least-Squares Quadratic Programming (F-SLSQP) and Attention-Guided Ensembles with Learnable Fuzzy Softmax to aggregate weight parameters from multiple models to capitalize on their collective knowledge and complementary representations. We perform statistical significance tests with 95% confidence intervals and $p$-values to analyze model performance. Our evaluations indicate models initialized with ImageNet-pretrained weights demonstrate superior generalizability over randomly initialized counterparts, contradicting some findings for non-medical images. Notably, ImageNet-pretrained models exhibit consistent performance during internal and external testing across different training scenarios. Weight-level ensembles of these models show significantly higher recall ($p<0.05$) during testing compared to individual models. Thus, our study accentuates the benefits of ImageNet-pretrained weight initialization, especially when used with weight-level ensembles, for creating robust and generalizable deep learning solutions.

## Author summary

In our research, we actively explore the optimal preparation of deep learning models for analyzing medical images. This area is of significant importance as medical images, such as chest X-rays (CXRs), differ from ordinary images and often present challenges for AI analysis. We experiment with three distinct initialization methods for preparing these

**Funding:** This study is supported by the Intramural Research Program (IRP) of the National Library of Medicine (NLM) and the National Institutes of Health (NIH).

**Competing interests:** The authors have declared that no competing interests exist.

models, and we make direct comparisons among them. Our aim centers around gauging the generalizability of our models trained on internal adult CXR data to the external CXR images collected from both adult and pediatric populations. Our findings indicate that models we initialize using a specific technique—known as "ImageNet-pretrained weight initialization"—generally outperform those we initialize randomly. This result offers a surprise, as existing research hints at different outcomes for non-medical images. Additionally, when we consolidate several of these models into an 'ensemble,' they achieve a more accurate identification of relevant cases during both internal and external testing. Therefore, our work underscores the promising potential of employing ImageNet-pretrained models and merging them into ensembles, aiming to enhance the reliability of AI in medical image analysis.

## Introduction

The prowess of Deep learning (DL) has been well established for medical imaging artificial intelligence (AI) applications with automation making way for improved and efficient image acquisition, quality assessment, object detection and tracking, disease screening, diagnostics, and prediction [1]. As a subset of machine learning (ML), DL comprises multilayered neural networks for automated feature extraction and predictions, outperforming traditional techniques in accuracy and robustness.

Chest X-rays (CXRs) are a routinely used diagnostic imaging modality. Despite lower sensitivity compared to computed tomography (CT) scans, CXRs offer several advantages, including cost-effectiveness, reduced radiation exposure, and accessibility, making them practical in resource-limited settings [2,3]. Several CXR datasets are available to the ML community which has resulted in significant advances in disease detection [4–8]. This dataset listing is not intended to be exhaustive as new datasets are being made available with higher frequency.

A key step in developing high-performing DL solutions is determining appropriate model initialization strategies [9]. Model initialization refers to the method of assigning initial values to neural network weights and biases. Optimal selection of the initialization strategy depends on various factors such as data characteristics including dimensionality, variability due to differences in patient anatomy, disease states, image acquisition procedures, and requirement for expert interpretation among others, activation functions, and optimization algorithms selected in the design [10]. Understanding the intricacies of model initialization and its impact on performance is essential for devising effective training methodologies, and addressing various issues in the training process, including vanishing or exploding gradients, achieving faster convergence, and stable training dynamics. An appropriately selected initialization strategy can also result in reliable and enhanced medical AI performance which is crucial for precision medicine applications.

The significance of model initialization is amplified when we consider challenges in model generalization which are primarily due to feature distribution shifts between training datasets and real-world use. For example, a model trained and tested on adult CXR data from the same source (*internal testing*) may result in significantly higher performance compared to testing it on adult CXR data from another source (*external testing*) [11]. Additional performance degradation may be observed when pediatric images exhibiting the same disease(s) are included in the testing. The inherent high-dimensional complexity and variability of medical images exacerbate this problem, causing models to overfit the training data. In this work, we present

findings from our investigations on the impact of different model initialization techniques on DL models and propose mechanisms to improve generalizability.

A review of DL literature on model initialization reveals two main techniques, namely, *cold-start* and *warm-start*, each with distinct implications for model training dynamics, generalizability, and performance [10]. The cold-start method initializes new weights and biases with small random values which results in training a new model from scratch. This technique offers an unbiased foundation but deprives the model of initialization guidance thereby resulting in slower convergence. Conversely, the warm-start strategy leverages weights and biases from a model that has been previously trained on data from similar content. Initialization guidance offered using this approach enables faster model convergence and also provides potentially enhanced performance. However, a previous study [10] has conversely reported that warm-start consistently underperforms with non-medical images, yielding models with poorer generalization and lower prediction accuracy compared to cold-start models. The *Shrink and Perturb* method proposed in [10] shrinks existing model weights towards zero and adds noise, resulting in faster training than cold-start and improved generalization over warm-start models. However, that and other studies focused on non-medical images [9,10,12–15] left a gap in understanding the impact of model initialization techniques on medical computer vision. Unlike non-medical images, medical images have unique characteristics including (i) variations in imaging modalities, e.g., CT, MRI, ultrasound, X-ray, pathology, endoscopy, where each modality captures different aspects of the human body at varying levels of resolution, contrast, and noise levels; (ii) image acquisition conditions including patient positioning, imaging protocols, and the expertise of medical professionals during acquisition that impacts the quality and appearance; (iii) varying anatomical structures that depict internal organs, tissues, and systems, and physiological processes that provide vital information for the diagnosis, treatment planning, and monitoring of diseases, (iv) limited and imbalanced data where instances of specific diseases or conditions with varying levels of progression are significantly smaller compared to healthy cases, and (v) ethical and regulatory considerations in handling medical data since they involve sensitive patient information, thereby ensuring the confidentiality and other critical factors [16,17].

Model generalizability is defined as the ability of a trained model to capture generalized patterns and perform well on unseen data [18]. Medical computer vision relies on model generalizability for several reasons [11] including accommodating patient diversity, adapting to various data sources and quality, addressing ethical considerations, and enhancing clinical utility. A general model is robust to different data sources and population distributions, considering factors such as the patient/study subject's ethnicity, sex, and severity of the disease(s) expressed on the image. Further, in many ML applications, data continuously flows into the system which may require regular model updates and may be unreasonable or difficult to implement. Therefore, developing reliable and generalizable models mandates both internal and external/out-of-distribution testing [19].

Most of the literature has focused on assessing internal generalization due to the lack of widely available data sets [5,20–22] and the findings, though significant, may not guarantee optimal model performance with external data. Federated learning methods have been proposed that use decentralized training to address challenges in achieving external generalization by incorporating diverse data distributions [23]. This approach could mitigate the risk of performance degradation when the model encounters unseen data distributions. However, this approach has its limitations, such as requiring consistent communication and synchronization between data sources, which can be challenging in real-world settings with privacy concerns or network instability [24]. Further, there could be data interoperability and completeness issues that limit generalization gains. Therefore, while federated learning provides a path

toward achieving external generalization, it also introduces new challenges. This presents us with an opportunity to consider and evaluate other novel and efficient methods for achieving external generalization.

For this work, we use adult and pediatric CXRs to evaluate model generalizability as they simultaneously exhibit significant similarities and differences due to anatomy and disease presentation across age groups [25]. These include: (i) Developmental stages: Evolving thoracic anatomy in pediatric patients is distinct in appearance from adults. There are thinner chest walls and more compliant rib cages in children. (ii) Unique abnormalities: Pediatric disease can present differently than adults or similar presentations could indicate different diseases. (iii) Imaging technique: Distinct protocols for pediatric CXRs can result in variations in intensity and contrast. Further, inspiration may be inconsistent across patients. (iv) Patient pose: Pediatric patients may need to be held down resulting in the presence of other hands in the image and unusual and variable pose of the patient. These discrepancies present challenges for DL models trained on adult data when directly applied to pediatric cases, potentially leading to sub-optimal generalizability, and reduced clinical utility. Prior work in pediatric CXR image analysis includes the development and evaluation of a ResNet-50 model trained to classify pediatric CXRs as showing pneumonia-consistent manifestations or normal lungs [11]. The model demonstrated comparatively improved performance on the internal test set (area under the curve (AUC): 0.95) compared to the external NIH-CXR test set (AUC: 0.54), highlighting potential limitations in model generalizability. There is limited literature analyzing the generalizability of deep models trained on adult CXRs to the pediatric population.

Our study presents key contributions to address the knowledge gap in the current literature regarding the impact of model initialization methods on the generalizability of DL models when we apply them to external adult and pediatric populations after training on internal adult CXR data. We specifically focus on scenarios with periodically arriving data for training, which is a common challenge faced by medical computer vision algorithms. Our investigation delves into the performance of widely used model initialization methods, providing insights into their adaptability and their implications on generalizability. Furthermore, we propose novel weight-level ensemble methods to improve model generalizability. This crucial understanding will pave the way for the successful deployment of DL models in medical imaging applications, ultimately improving clinical decision-making and patient outcomes.

## Materials and methods

Datasets

This retrospective study utilizes the following datasets:

i. RSNA-CXR dataset: This publicly available CXR collection results from a collaboration between the RSNA, the Society of Thoracic Radiology (STR), and the National Institutes of Health (NIH) for the Kaggle pneumonia detection challenge [26]. The objective was to help support the design and development of image analysis and ML algorithms through a challenge targeting automatic classification of CXRs as normal, containing non-pneumonia-related, or pneumonia-related opacities. The collection comprises 26,684 deidentified anterior-posterior (AP) and posterior-anterior (PA) CXRs in DICOM format, featuring 8,851 normal lungs and 17,833 other abnormal radiographic patterns, of which 6,012 manifest pneumonia-related opacities. We use this dataset to train, validate, and internally test the DL model.

ii. Indiana-CXR dataset: The Indiana CXR dataset contains 7,470 frontal and lateral CXR projections [27] in DICOM format, accompanied by multiple annotations, including

indications, findings, and impressions in textual form. These images are sourced from hospitals affiliated with the Indiana University School of Medicine. Among these, 2,378 PA CXRs exhibit abnormal pulmonary manifestations, and 1,726 CXRs have normal lung appearances. This de-identified dataset is stored at the National Library of Medicine (NLM) and has been exempted from Institutional Review Board review (OHSRP # 5357). We use this dataset as the external adult test set.

iii. VINDR-PCXR dataset: The VINDR-PCXR dataset is a publicly available pediatric CXR collection [28] developed to support computer-aided diagnosis algorithm development for pediatric CXR interpretation. It consists of 9,125 CXR scans, in DICOM format, collected from three major Vietnamese hospitals between 2020 and 2021. The pediatric dataset includes deidentified images of 5,354 males, 3,709 females, and 62 patients with unknown gender. Among the 8,755 pediatric CXRs, 5,876 show normal lungs, and 2,879 exhibit other cardiopulmonary abnormalities, with age distributions as follows: 5,335 CXRs for ages 1 day to under 24 months, 3,351 CXRs for ages 24 months to under 11 years, and 69 CXRs for ages 11 to under 18 years. We use this dataset as an external pediatric test.

iv. NIH-CXR dataset: The NIH-CXR dataset is a publicly accessible, large-scale collection of deidentified CXRs [29] compiled by the NIH Clinical Center. It contains 112,120 frontal-view CXR images in PNG format, from 30,805 unique patients. The dataset includes 14 cardiopulmonary disease labels, text-mined from radiological reports using a Natural Language Processing (NLP) labeler. Among these, 5,257 pediatric CXRs represent normal lungs (n = 3,066) and other cardiopulmonary abnormalities (n = 2,191), divided into three age groups: 34 CXRs captured from pediatric patients of ages 1 day to under 24 months, 1,787 CXRs of ages 24 months to under 11 years, and 3,486 CXRs of ages 11 to under 18 years. The pediatric group consists of 3,018 males and 2,239 females, while 106,863 CXRs belong to patients older than 18 years. We use this dataset as the external pediatric test.

We further partition the RSNA-CXR dataset at the patient level into 70% for training, 10% for validation, and 20% for internal testing. The training and validation sets are additionally divided into two equal-sized subsets to simulate periodic data arrival for training and validation and facilitate the simplest case of warm-start. The DL model trains to converge on the first half of the data and then trains on the full collection, which represents 100% of the data. We name the first half *RSNA-Partial (P)* and the full collection *RSNA-Full (F)*. The internal test remains the same for both the RSNA-P and RSNA-F datasets. Table 1 provides details of this partition.

The external test sets consist of adult CXRs from the Indiana-CXR collection and pediatric CXRs from the NIH-CXR and VINDR-PCXR collections. We categorize the pediatric CXRs into three groups: Ped-2 (1 day to under 24 months), Ped-11 (24 months to under 11 years), and Ped-18 (11 years to under 18 years), based on the lung developmental stages from infancy to adulthood as discussed in [30]. Table 2 shows the categorization of test CXRs according to various age groups.

**Table 1. Training, validation, and internal test split using the RSNA-CXR dataset.**

| Dataset | Train | | Val | | Internal test | |
|---------|-------|---|-----|---|--------------|---|
| | No finding | Abnormal | No finding | Abnormal | No finding | Abnormal |
| RSNA-P | 3098 | 6242 | 442 | 891 | 1770 | 3566 |
| RSNA-F | 6196 | 12484 | 885 | 1783 | 1770 | 3566 |

**Table 2. External test set categorization across various age groups.**

| Dataset | 1 day to < 24 months | | 24 months to < 11 years | | 11 years to < 18 years | | > 18 years | |
|---|---|---|---|---|---|---|---|---|
| | No finding | Abnormal | No finding | Abnormal | No finding | Abnormal | No finding | Abnormal |
| Indiana-CXR | - | - | - | - | - | - | 1726 | 2378 |
| NIH-CXR | 29 | 5 | 1059 | 728 | 1978 | 1458 | - | - |
| VINDR-PCXR | 3341 | 1994 | 2475 | 876 | 60 | 9 | - | - |
| **Total** | **3370** | **1999** | **3534** | **1604** | **2038** | **1467** | **1726** | **2378** |

## Lung region delineation and cropping

We utilize a UNet [31] model with an ImageNet-pretrained Inception-V3 encoder backbone from our previous study [32] to delineate the lung regions and crop them to the size of a bounding box. The purpose of lung cropping is to prevent the DL model from learning irrelevant features for cardiopulmonary disease detection. We resize the cropped lung bounding boxes to 256×256-pixel dimensions and normalize them to the range [0, 1] to reduce computational complexity.

## Model architecture and training

For the model architecture and training scenario, we employ a VGG-16 model [33] architecture. We truncate it at its deepest pooling layer and append a global average pooling (GAP) layer and a final dense layer with two nodes and Softmax activation. This modified model, referred to as *VGG-16-M*, predicts whether the CXRs show normal lungs or other cardiopulmonary abnormalities. We chose the VGG-16 model due to its simplicity, effectiveness, and well-known performance in medical image classification tasks, particularly using CXRs [34–36]. Selecting an optimal model falls beyond the scope of this research, as our study aims to analyze the impact of model initialization strategies on deep model generalization. The proposed technique can be applied to any model suitable for the characteristics of the data under study. Table 3 provides a list of the data and model terminologies used in this study.

Each model undergoes training and validation using the RSNA-CXR dataset with a mini-batch size of 64. We utilize the Adam optimizer with an initial learning rate of 0.001 to minimize the categorical cross-entropy loss. Model checkpoints are stored via callbacks when a decrease in validation loss is observed. The checkpoint exhibiting the lowest validation loss is used to generate predictions for both the internal and external test datasets. Test performance evaluation occurs at the ideal classification threshold, determined by maximizing the F-score for the validation dataset.

**Table 3. Data and model terminologies.**

| Terminologies | Explanation |
|---|---|
| R, I | Model initialization: random weights (R) or ImageNet-pretrained weights (I) |
| P, F | VGG-16-M model dataset usage: RSNA-P (P) or RSNA-F (F) |
| Cold-RP | Random initialization, trained on RSNA-P |
| Cold-IP | ImageNet-pretrained initialization, trained on RSNA-P |
| Cold-RF | Random initialization, trained on RSNA-F |
| Warm-RF | Cold-RP model fine-tuned on RSNA-F |
| Shrink-RF | Cold-RP model with weights shrunk by factor $\alpha1$, found via Bayesian search |
| Cold-IF | ImageNet-pretrained initialization, trained on RSNA-F |
| Warm-IF | Cold-IP model fine-tuned on RSNA-F |
| Shrink-IF | Cold-IP model with weights shrunk by factor $\alpha2$, found via Bayesian search |

## Optimizing the weight-scaling factor

The method proposed in the Shrink and Perturb technique [10] involves shrinking the existing model weights by multiplying with a factor $\alpha$ and incorporating a small noise $\beta$ to accelerate DL model convergence and enhance generalization compared to standard cold-start and warm-start methods. Let $W$ be the set of model weights. We calculate the updated weights $W'$ using Eq (1):

$$W' = \alpha W + \beta. \tag{1}$$

Here, $\alpha$ denotes the weight-scaling factor. Previous experiments [10] used discrete $\alpha$ values and fixed $\beta$ at 0.01. In contrast, while we continue to use a fixed value for $\beta$ as 0.01, we apply Bayesian optimization via Gaussian Process (GP) minimization [37] to identify the optimal $\alpha$ for shrinking the weights of the Cold-RP and Cold-IP models. These are subsequently used to initialize the weights in the Shrink-RF and Shrink-IF models, respectively. Bayesian optimization using GP minimization reduces susceptibility to local minima, enabling more effective identification of the optimal $\alpha$ within a continuous interval compared to the grid or random search methods at discrete intervals. GP minimization explores the search space more thoroughly and converges efficiently by modeling the objective function as a Gaussian process sample. We define the continuous search space for $\alpha$ within the range [0.1, 0.9]. We create a function that accepts $\alpha$ as input and performs the following steps: (i) instantiate and compile the model with the current weights, (ii) train and validate the model, storing the best model weights, validation loss, $\alpha$, and training history whenever the validation loss decreases, and (iii) perform GP minimization for 100 function calls and 30 random starts to converge to the optimal $\alpha$ with minimal validation loss. The hyperparameters for GP minimization follow the default settings in the scikit-optimize Python library.

## Weight-level ensembles

We are also proposing ensemble methods that merge the weights of multiple models. Our approach is different from traditional techniques that aggregate model predictions [38–41]. Our proposed *weight-level ensembles* harness the power of diverse weight initializations, capitalizing on complementary learning dynamics to foster robust generalization in complex, high-dimensional medical data landscapes.

We perform Equal Weight Averaging (EWA), which combines the weights of multiple trained models to create an average model. This technique aims to enhance classification performance by leveraging the complementary strengths of individual models in capturing data patterns. We achieve this by iterating through each model's layers, retrieving and averaging the layer weights with equal weight factors, resulting in a new model with a similar architecture for prediction.

We introduce a novel F-score-weighted Sequential Least-Squares Quadratic Programming (F-SLSQP)-based weighted ensemble method to determine the optimal multiplication factors for combining the weights of multiple models in the ensemble. We identify these optimal factors by minimizing the error, as defined in Eq (2), through SLSQP-based constrained minimization [42].

$$Error = 1 - (F-score_{validation}). \tag{2}$$

The process of determining the optimal multiplication factors involves the following steps: (i) defining a function to compute the weighted average of weights for the ensemble models, (ii) defining a function to create a new model with the same architecture as the models in the ensemble, (iii) creating a global variable for the best multiplication factors, (iv) defining a

function to calculate the error from the weighted average of the models, (v) setting the optimization parameters, including the constraints and bounds, where the constraint ensures the sum of scaling factors equals 1.0 and the bounds ensure each scaling factor is within the range [0, 1], (vi) executing the SLSQP algorithm multiple times (n = 100) to minimize the error and find the optimal multiplication factors, (vii) performing weighted averaging with the optimal multiplication factors to create the weighted ensemble model, and (viii) compiling and saving the weighted ensemble model for prediction.

Additionally, we present a novel method for developing an attention-guided ensemble incorporating a learnable Fuzzy Softmax layer (AGELFS). This technique utilizes attention mechanisms [43] to emphasize relevant features of each model while mitigating less significant ones. The ensemble construction involves the following steps: (i) instantiating and freezing the constituent models with their respective weights, (ii) processing training input through these models and appending a GAP layer to each model's output, (iii) concatenating the outputs of the GAP layers, (iv) introducing an attention layer to derive attention-based weights for the concatenated outputs, (v) appending a dense layer with conventional Softmax activation to learn attention-based weights for the concatenated outputs, (vi) applying a learnable Fuzzy Softmax (LFS) layer to the dense layer output, and (vii) training the ensemble. The Fuzzy Softmax layer [44] enhances the conventional Softmax function by introducing a learnable Fuzziness parameter that controls the uncertainty level in output probabilities, as described in Eq (3), where $x\_i$ and $x\_j$ represent the input logits, and *Fuzziness* is the learnable parameter.

$$(Learnable\ Fuzzy\ Softmax(x\_i) = \exp(fuzziness * x\_i)) / sum(\exp(fuzziness * x\_j)). \tag{3}$$

## Performance evaluation and statistical significance analysis

We examine model performance using key metrics, including balanced accuracy, precision, recall, the area under the precision-recall curve (AUPRC), F-score, and Matthews Correlation Coefficient (MCC). Each metric provides valuable insights into the model's effectiveness in various aspects of classification tasks. The Null Hypothesis ($H_0$) is that there is no statistically significant difference in performance between the different weight initialization models compared, as measured by the Matthews Correlation Coefficient (MCC). The Alternate Hypothesis ($H_1$) is that there is a statistically significant difference in performance between the models compared. We present the statistical significance of the MCC by utilizing 95% binomial confidence intervals (CIs) and ascertain them through the Clopper-Pearson Exact methodology to distinguish model efficacy. We determine the *p*-values based on the CI-based Z-test [45]. We obtain the MCC values and their corresponding 95% CIs for the compared models. For each model, we compute the standard error (SE) using Eq (4):

$$SE = (CI_{upper} - CI_{lower}) / (2 * 1.96). \tag{4}$$

Here, $CI_{upper}$ and $CI_{lower}$ represent the upper and lower bounds of the CIs, respectively. We compute the difference in the MCC (*ΔMCC*) and SE (*ΔSE*) values using Eqs (5) and (6) respectively:

$$\Delta MCC = MCC2 - MCC1. \tag{5}$$

$$\Delta SE = sqrt(SE1^2 + SE2^2). \tag{6}$$

Here, *MCC1*, *MCC2*, *SE1*, and *SE2* are the MCC and SE metrics of the compared models. We compute the Z-score from this difference using Eq (7):

$$Z = \Delta MCC / \Delta SE. \tag{7}$$

We calculate the corresponding *p*-value for the Z-score using an online Z-table. A threshold of 0.05 is utilized to establish statistical significance using the 95% CIs. If the *p*-value is less than 0.05, we conclude that the difference in performance is statistically significant, thereby rejecting $H_0$ in favor of $H_1$. We apply a similar methodology to evaluate the statistical significance of recall metrics for the proposed weight-level ensembles.

## Results and discussion

We first present a comparative analysis between the performances of the Cold-RP and Cold-IP models. Recall that the Cold-RP model initializes the VGG-16 backbone of the VGG-16-M model with random weights and trains it on the RSNA-P dataset. Conversely, the Cold-IP model initializes the VGG-16 backbone of the VGG-16-M model with ImageNet-pretrained weights and also trains it on the RSNA-P dataset.

Table 4 displays performance metrics when predicting the RSNA-P test set (internal adult test set), while Fig 1 illustrates the AUPRC, confusion matrices, and a comparison of MCC values. Based on the information in Table 4, we deduce the following: (i) The Cold-IP model converges considerably faster than the Cold-RP model, and (ii) the Cold-IP model exhibits a significantly higher MCC ($p<0.00001$) and notably higher values for other performance metrics compared to the Cold-RP model.

The terms B. Acc., P, R, and F denote balanced accuracy, precision, recall, and F-score, respectively. Bold numerical values denote superior performance in respective columns. Values in parentheses represent the 95% CIs for the MCC metric. The * denotes statistically significant MCC ($p<0.00001$).

Fig 2 depicts histograms that illustrate the distribution of Softmax activations for the positive (1—*Abnormal*) and negative (0—*No Finding*) classes when predicting the RSNA-P test set using the Cold-RP and Cold-IP models. The Softmax histograms provide insight into the correctness and confidence of each model's predictions, as well as differences in Softmax predictions and overall performance. The x-axis represents Softmax activations, and the y-axis indicates the density of these activations. The histograms' shape and density reveal a more distinct separation between the two classes in the Cold-IP model, characterized by two clear peaks near 0 and 1. This distinction may result from the Cold-IP model's initialization with ImageNet-pretrained weights, allowing it to leverage useful features learned from a large-scale dataset.

Consequently, the model converges more effectively and generates more accurate and confident predictions for both classes. In contrast, the Cold-RP model, initialized with random weights, exhibits a less distinct separation and a wider distribution of predictions around 0.5, suggesting lower confidence and correctness in its predictions. These findings underscore the superior performance of the Cold-IP model relative to the Cold-RP model.

We also use t-SNE visualizations [46] to assess the feature representations learned by the Cold-RP and Cold-IP models in the 2D space (Fig 3). The t-SNE visualization allows us to

**Table 4. Performance of models initialized with random and ImageNet-pretrained weights on the internal adult test set.**

| Model | AUPRC | B. Acc. | P | R | F | MCC | Training time (in sec.) | *p*-MCC |
|-------|-------|---------|---|---|---|-----|-------------------------|---------|
| Cold-RP | 0.9452 | 0.8156 | 0.8856 | 0.8531 | 0.8690 | 0.6204 (0.6073, 0.6335) | 2052.83 | <0.00001 |
| Cold-IP | **0.9671** | **0.8466** | **0.8961** | **0.9044** | **0.9002** | **0.6964 (0.6840, 0.7088)*** | **812.52** | |

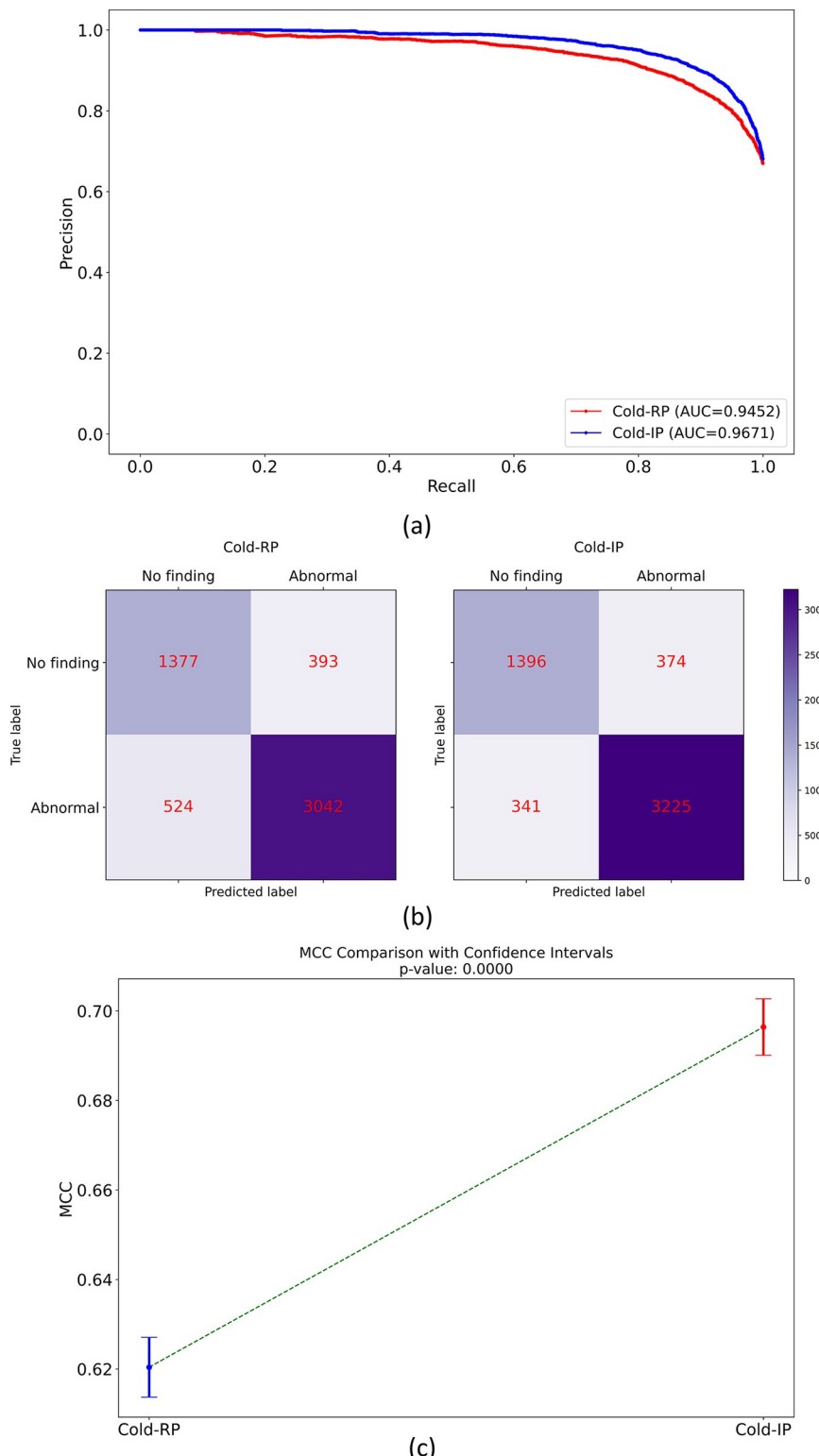

**Fig 1. Internal adult test performance comparison between the Cold-RP and Cold-IP models.** (a) AUPRC, (b) Confusion matrices, and (c) MCC comparison with the *p*-value.

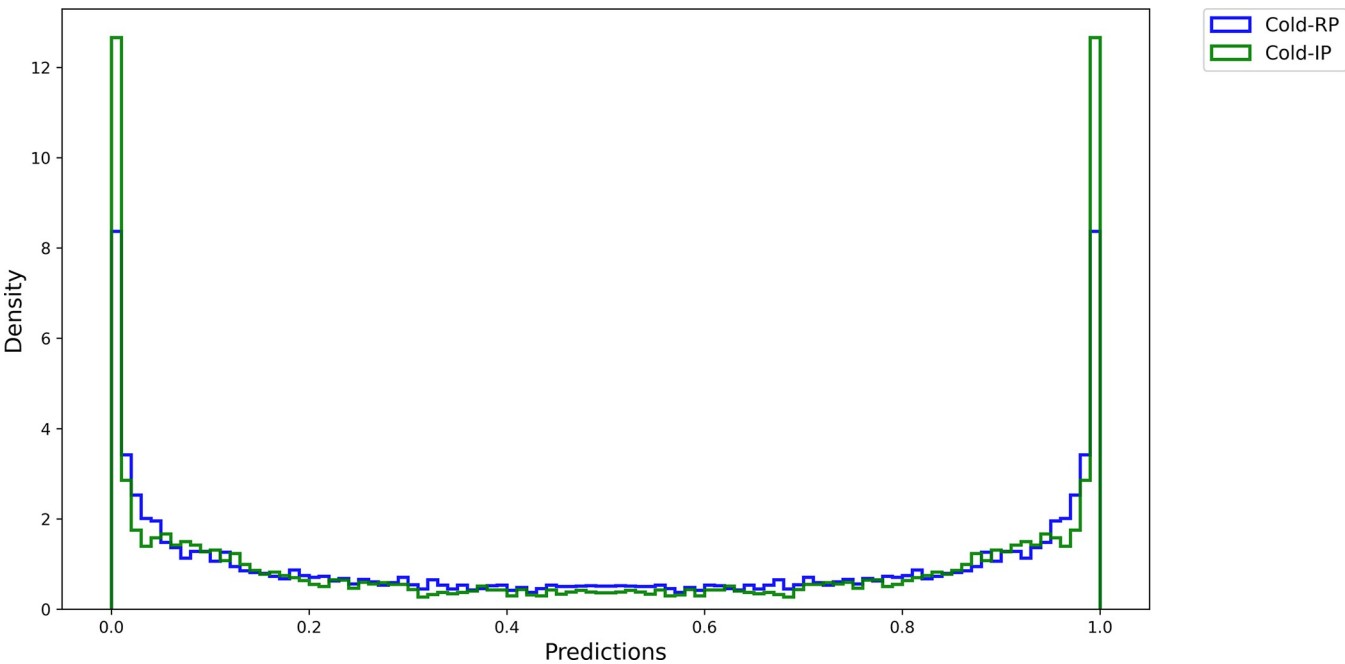

**Fig 2. Histograms of the Softmax activations of Cold-RP and Cold-IP models.**

effectively evaluate each model's ability to capture the data's underlying structure and its generalizability. We determine the optimal perplexity and learning rate parameters through rigorous empirical analysis. The t-SNE plot highlights distinct visual disparities in the models' learned features. Although both models acquire meaningful data representations, the Cold-IP model's t-SNE presents two well-defined clusters for the *No Finding* and *Abnormal* classes, indicating that the Cold-IP model more effectively captures the data's essential features and

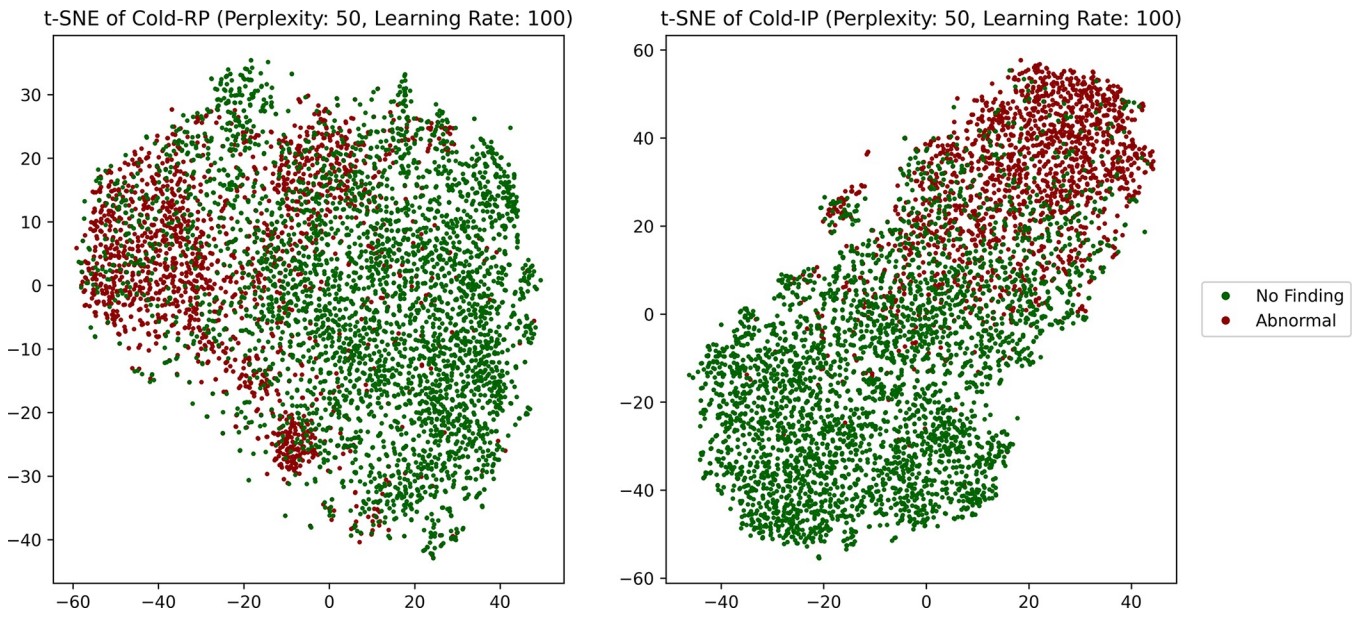

**Fig 3. The t-SNE visualization of the features learned by the Cold-RP (left) and Cold-IP (right) models.**

**Table 5. Performance of models on the internal adult test set.**

| Model | AUPRC | B. Acc. | P | R | F | MCC | Training time (in sec.) | *p*-MCC |
|---|---|---|---|---|---|---|---|---|
| Cold-RF | 0.9557 | 0.8383 | 0.9015 | 0.8676 | 0.8842 | 0.6650 (0.6523,0.6777) | 3098.99 | <0.00001 |
| Cold-IF | **0.9732** | **0.8534** | 0.8958 | **0.9232** | **0.9093** | **0.7187 (0.7066,0.7308)** * | **1458.36** | |
| Warm-RF | 0.9522 | 0.8036 | 0.8593 | 0.9061 | 0.8821 | 0.6267 (0.6137,0.6397) | 1453.58 | <0.00001 |
| Warm-IF | **0.9723** | **0.8686** | **0.9214** | 0.8904 | **0.9056** | **0.7258 (0.7138,0.7378)** * | **1067.58** | |
| Shrink-RF | 0.9572 | 0.8158 | 0.8711 | 0.8999 | 0.8853 | 0.6431 (0.6302,0.6560) | 1982.63 | <0.00001 |
| Shrink-IF | **0.9714** | **0.8508** | **0.8934** | **0.9237** | **0.9083** | **0.7150 (0.7028,0.7272)** * | **1205.31** | |

Bold numerical values denote superior performance in their respective columns. The

* denotes statistically significant MCC among each model pair, i.e., (Cold-RF, Cold-IF), (Warm-RF, Warm-IF), and (Shrink-RF, Shrink-IF) ($p<0.00001$).

generalizes to the internal adult test set. This representation can potentially enhance classification performance on unseen data. Conversely, the Cold-RP model's t-SNE displays greater class overlap. Despite some separation, the clusters are less defined compared to the Cold-IP model. This diminished separation implies that the Cold-RP model struggles to accurately classify test set instances, particularly near the decision boundary. The t-SNE visualizations underscore that the Cold-IP model exhibits superior generalization capabilities relative to the Cold-RP model.

We proceed to train and evaluate models on 100% of the data, i.e., the RSNA-F dataset, with the aforementioned configurations for Cold-RF, Warm-RF, Shrink-RF, Cold-IF, Warm-IF, and Shrink-IF models (Table 3). The weights of the Cold-RP and Cold-IP models that are used to initialize the weights for the Shrink-RF model and the Shrink-IF models, respectively, are shrunk by an optimal scaling factor of 0.7209 ($\alpha1$) and 0.9 ($\alpha2$), respectively, as determined by Bayesian optimization through GP minimization in the constrained continuous interval of [0.1, 0.9].

Table 5 displays performance metrics, while Fig 4 illustrates the AUPRC achieved by each model when predicting the RSNA-F test (i.e., the internal adult test set). We observe that the models initialized with ImageNet-pretrained weights (Cold-IF, Warm-IF, Shrink-IF) converge considerably faster and also significantly outperform their randomly initialized counterparts (Cold-RF, Warm-RF, Shrink-RF) in terms of MCC ($p<0.00001$) and other metrics.

Among the ImageNet-initialized models, the Cold-IF model takes slightly longer to converge (1458.36 seconds) compared to the Warm-IF (1067.58 seconds) and Shrink-IF models (1205.31 seconds). The confusion matrices of these models are shown in S1 Fig. S2 Fig shows a comparison of MCC values for each model pair, i.e., (Cold-RF, Cold-IF), (Warm-RF, Warm-IF), and (Shrink-RF, Shrink-IF).

Analyzing the t-SNE visualizations in Fig 5 allows us to glean insights into the generalization abilities of each model within their respective pairs: (Cold-RF, Cold-IF), (Warm-RF, Warm-IF), and (Shrink-RF, Shrink-IF). We determine the ideal perplexity and learning rate values for each model through extensive empirical evaluations. In the Cold-RF versus Cold-IF comparison, the Cold-IF model, initialized with ImageNet-pretrained weights, showcases more distinct clustering and superior class separation than its randomly initialized counterpart. Similarly, the Warm-IF model demonstrates clearer data point separation into distinct clusters compared to Warm-RF in their respective comparison. Lastly, the Shrink-IF model presents more well-defined clusters and class separations than the Shrink-RF. Based on the t-SNE visualizations, we observe that ImageNet-initialized models (Cold-IF, Warm-IF, Shrink-IF) exhibit enhanced generalization capabilities compared to their randomly initialized counterparts. These observations underscore the significance of employing ImageNet-pretrained weights to boost performance and generalizability in such models.

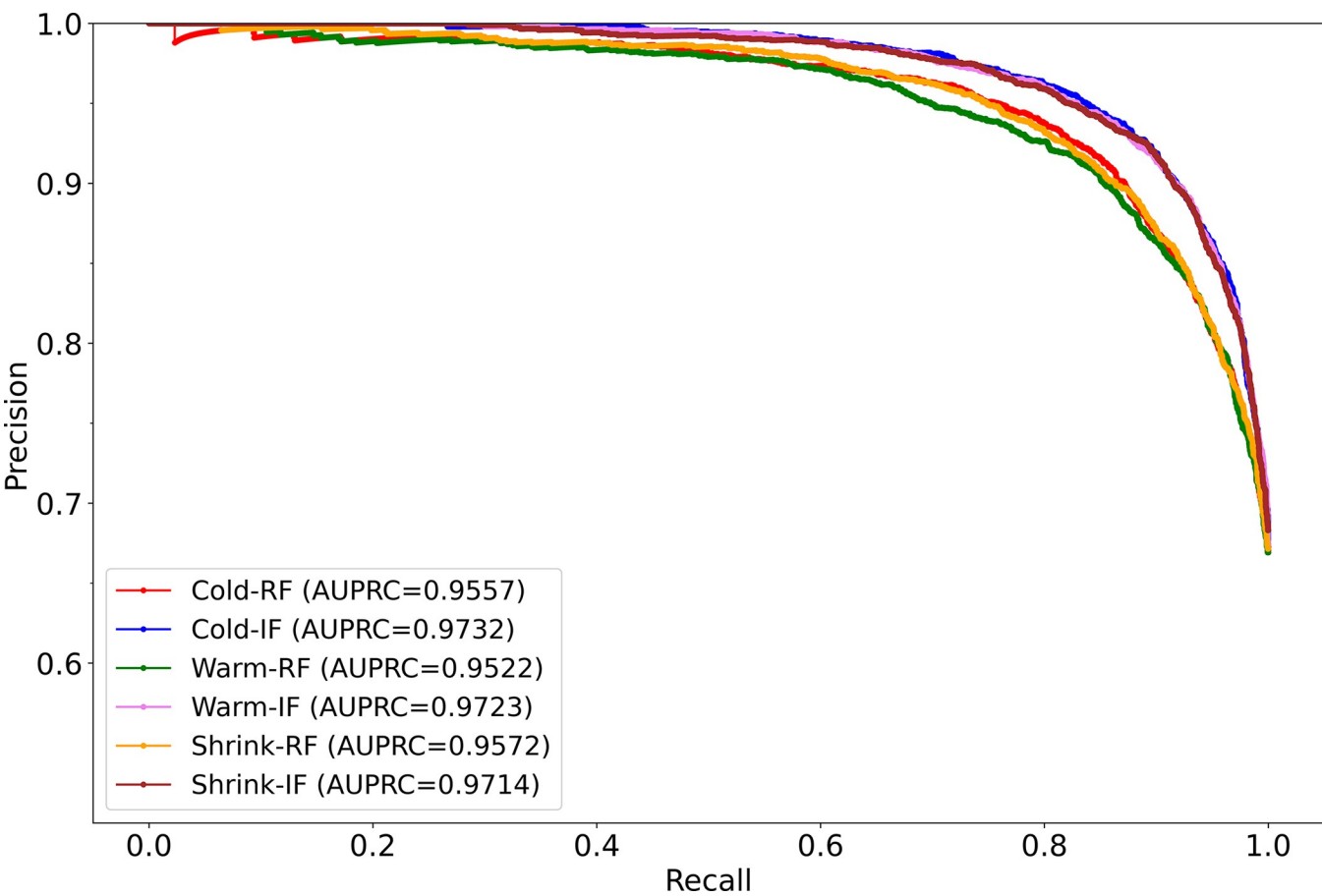

**Fig 4. AUPRC of the models while predicting the internal adult test set.**

The Softmax activation histograms (S3 Fig) help visualize performance disparities. In the Cold-RF versus Cold-IF comparison, the Cold-IF model exhibits a bimodal distribution with peaks near 0 and 1, suggesting confident, accurate predictions for both classes. Conversely, the Cold-RF model displays a uniform distribution without a preference for either class, indicating less confident, less accurate predictions. In the Warm-RF versus Warm-IF comparison, we observe that the Warm-IF model's histogram displays a distinct bimodal distribution, indicative of confident, accurate predictions. The Warm-RF model exhibits a less pronounced bimodal distribution, signaling lower prediction confidence. The Warm-IF model's superior performance corresponds with its histogram's well-defined bimodal distribution. Similarly, the Shrink-RF and Shrink-IF model pair reveal performance differences. The Shrink-IF model's histogram presents a prominent bimodal distribution, implying greater confidence and accuracy in predictions, whereas the Shrink-RF model shows a less distinct distribution, reflecting weaker prediction capabilities.

As we examine Table 5, we make an intriguing observation. When predicting the internal adult test, the Cold-IF model only marginally outperforms the Warm-IF and Shrink-IF models in terms of AUPRC, F-score, and MCC. The Warm-IF model shows slightly higher balanced accuracy and precision, while the Shrink-IF model exhibits marginally better recall. However, there are no significant performance differences observed for the MCC metric ($p > 0.05$). Other metrics also demonstrate similar values across the models. Nevertheless, considering

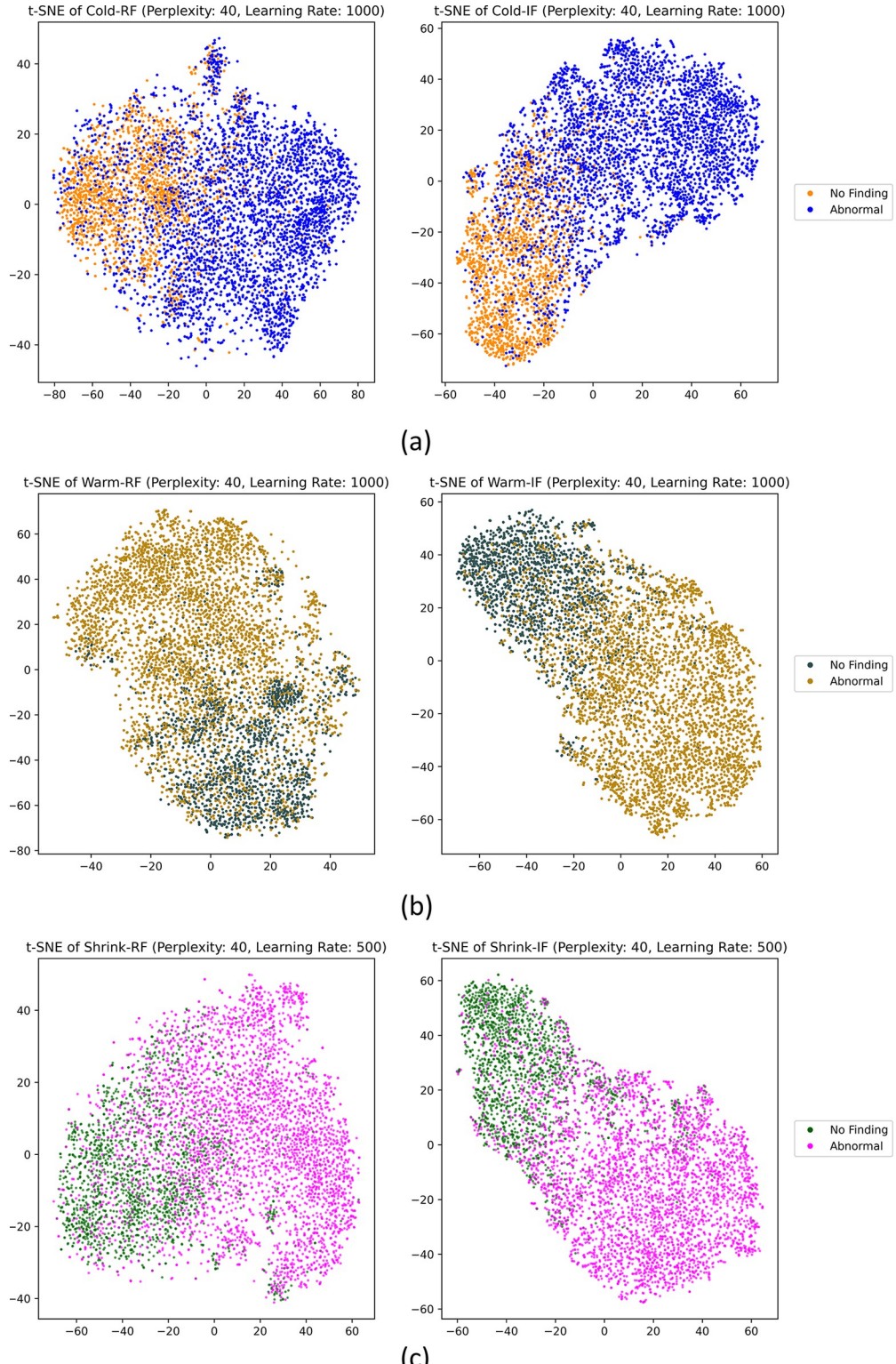

**Fig 5. t-SNE visualization of the features learned by the model pairs.** (a) Cold-RF and Cold-IF; (b) Warm-RF and Warm-IF, and (c) Shrink-RF and Shrink-IF.

**Table 6. Comparing the model performances when predicting the external adult and pediatric test sets.** Bold numerical values denote superior performance in their respective columns.

| Test | Models | AUPRC | B. Acc. | P | R | F | MCC | *p*-MCC |
|------|--------|-------|---------|---|---|---|-----|---------|
| Adult | Cold-IF | 0.8490 | **0.7170** | 0.8452 | **0.5807** | **0.6884** | **0.4378 (0.4226, 0.4530)** | >0.05 |
| | Warm-IF | **0.8573** | 0.6942 | **0.8939** | 0.4643 | 0.6112 | 0.4180 (0.4029, 0.4331) | |
| | Shrink-IF | 0.8472 | 0.7073 | 0.8599 | 0.5345 | 0.6592 | 0.4263 (0.4111, 0.4415) | |
| Ped-2 | Cold-IF | **0.4685** | **0.5480** | **0.3997** | **0.8794** | **0.5496** | **0.1206 (0.1118, 0.1294)** | >0.05 |
| | Warm-IF | 0.4289 | 0.5394 | 0.3953 | 0.8514 | 0.5399 | 0.0955 (0.0876, 0.1034) | |
| | Shrink-IF | 0.4589 | 0.5362 | 0.3927 | 0.8769 | 0.5425 | 0.0936 (0.0858, 0.1014) | |
| Ped-11 | Cold-IF | **0.5936** | 0.6235 | 0.4861 | **0.7519** | **0.5905** | 0.2458 (0.2340, 0.2576) | >0.05 |
| | Warm-IF | 0.5876 | **0.6327** | **0.5063** | 0.6976 | 0.5868 | **0.2595 (0.2475, 0.2715)** | |
| | Shrink-IF | 0.5887 | 0.6243 | 0.4881 | 0.7444 | 0.5896 | 0.2465 (0.2347, 0.2583) | |
| Ped-18 | Cold-IF | 0.6726 | 0.7116 | 0.5871 | 0.8569 | 0.6968 | 0.4281 (0.4117, 0.4281) | >0.05 |
| | Warm-IF | 0.682 | **0.7324** | **0.6229** | 0.8241 | **0.7095** | **0.4614 (0.4448, 0.4780)** | |
| | Shrink-IF | **0.6822** | 0.7113 | 0.5849 | **0.8643** | 0.6977 | 0.4293 (0.4129, 0.4457) | |

their superior performance compared to their randomly initialized counterparts, the Cold-IF, Warm-IF, and Shrink-IF models did not demonstrate significant differences in their generalizability to the internal test set.

Similar trends are observed when assessing external generalization in Table 6. For the external adult test, the Cold-IF model only marginally, but not significantly ($p>0.05$), outperforms the Warm-IF and Shrink-IF models in terms of MCC. This observation holds for balanced accuracy, recall, and F-score. The Warm-IF model slightly outperforms the other models in terms of AUPRC and precision. When predicting the external Ped-2 test, the Cold-IF model slightly outperforms the Warm-IF and Shrink-IF models in terms of all metrics. However, no significant difference in performance is observed for the MCC metric ($p>0.05$). Similar performance trends are observed for the Ped-11 and Ped-18 test sets. With the Ped-11 test, the Warm-IF model marginally outperforms ($p>0.05$) the other models in terms of MCC. The Cold-IF model demonstrates slightly superior values for AUPRC, recall, and F-score. The Shrink-IF model performs the worst among all models. With the Ped-18 test, the Warm-IF model achieves marginally superior balanced accuracy, precision, F-score, and MCC. The Shrink-IF model shows slightly better recall and AUPRC, while the Cold-IF model exhibits the lowest performance. These observations suggest that, despite differences in training scenarios, the ImageNet-initialized models, namely Cold-IF, Warm-IF, and Shrink-IF, might have converged to distinct local optima that enable comparable generalization performance across the external test sets.

To assess weight distribution similarity, we generated a heatmap of Earth Mover Distance (EMD) values in Fig 6. Lower EMD values indicate higher weight similarity due to shared ImageNet-pretrained weight initialization for the Cold-IF, Warm-IF, and Shrink-IF models. This similarity, supported by the low EMD values, aligns with the observation that the models' performance differences are not pronounced.

We further analyze the weight distribution similarity of the Cold-IF, Warm-IF, and Shrink-IF models using scatter plots (Fig 7). The plots visually depict the relationship between the weight distributions of each model pair, namely (Cold-IF, Warm-IF), (Cold-IF, Shrink-IF), and (Warm-IF, Shrink-IF). Each point in the scatter plot represents a pair of weights from the compared models, with the x-axis and y-axis representing the weights of the respective models. Dense point distributions along the diagonal indicate higher weight similarity, while more dispersed distributions suggest less similarity. We observe a strong positive correlation between

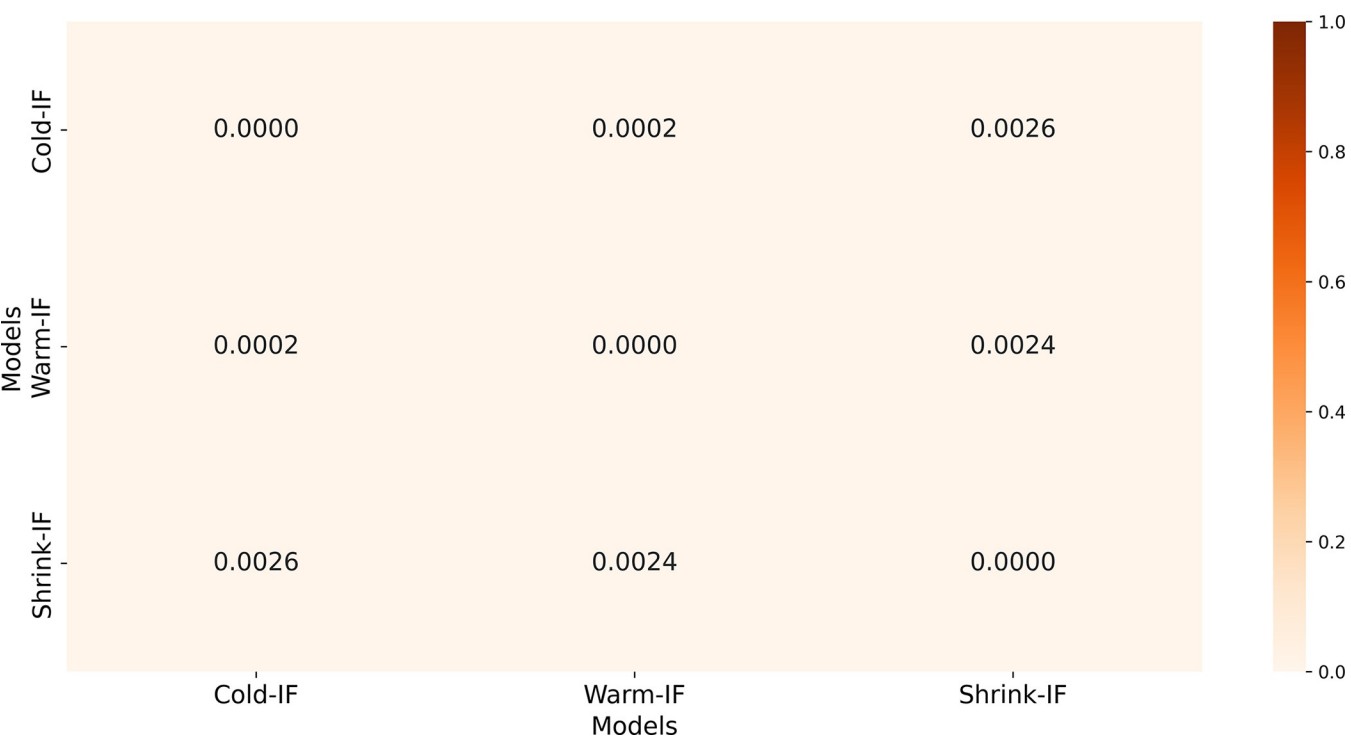

**Fig 6. Heatmap showing EMD values between each model pair for the Cold-IF, Warm-IF, and Shrink-IF models.**

weight distributions as evident from the scatter plot patterns. The scatter plots demonstrate a dense diagonal distribution, indicating highly similar weight distributions for the compared models. This similarity implies that the models learned similar features and representations during training, resulting in comparable Softmax predictions for the positive and negative classes, as supported by their performance metrics.

We also apply ensemble methods to evaluate if the generalization performance with internal and external test sets can surpass that of the individual models. Table 7 presents the ensemble performances when predicting the internal adult test.

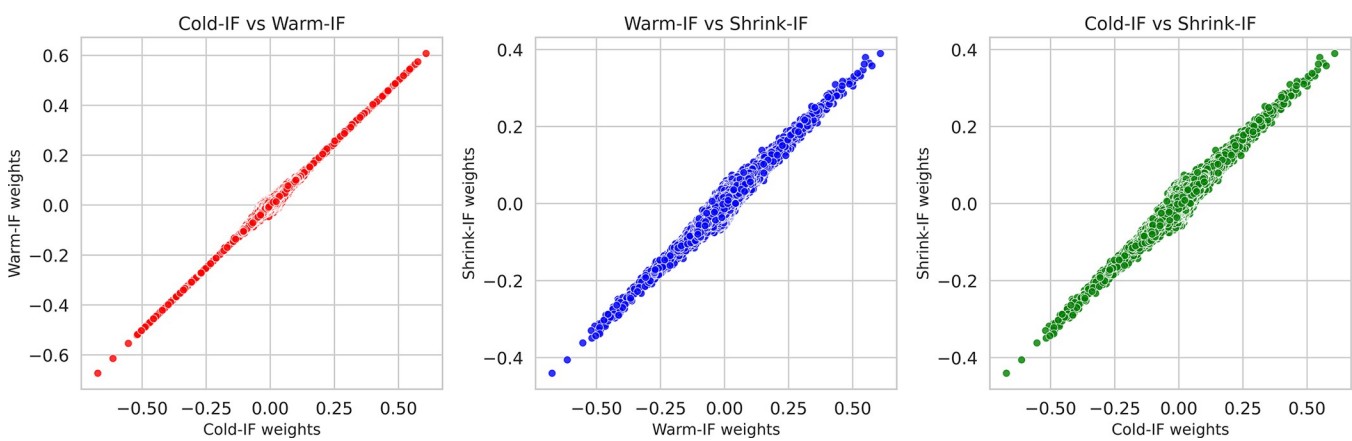

**Fig 7. Scatter plots show the correlation in weights between pairs of models.**

**Table 7. Model performances achieved with the internal adult test.**

| Models | AUPRC | B. Acc. | P | R | F | MCC |
|---|---|---|---|---|---|---|
| Warm-IF-Baseline | 0.9723 | 0.8686 | 0.9214 | 0.8904 | 0.9056 | 0.7258 (0.7138,0.7378) |
| **EWA Ensemble** | | | | | | |
| Cold-IF, Warm-IF | 0.9709 | 0.8548 | 0.8999 | 0.9148 | 0.9073 | 0.7159 (0.7037,0.7281) |
| Cold-IF, Shrink-IF | 0.9707 | 0.8611 | 0.9100 | 0.9019 | 0.9059 | 0.7190 (0.7069,0.7311) |
| Warm-IF, Shrink-IF | 0.9730 | 0.8697 | 0.9200 | 0.8965 | 0.9081 | **0.7305 (0.7185,0.7425)** |
| Cold-IF, Warm-IF, Shrink-IF | 0.9712 | 0.8551 | 0.8993 | 0.9170 | 0.9081 | 0.7177 (0.7056,0.7298) |
| **F-SLSQP Ensemble** | | | | | | |
| Cold-IF, Warm-IF | 0.9724 | 0.8680 | 0.9204 | 0.8915 | 0.9057 | 0.7254 (0.7134,0.7374) |
| Cold-IF, Shrink-IF | 0.9722 | 0.8635 | 0.9096 | 0.9089 | 0.9092 | 0.7266 (0.7146,0.7386) |
| Warm-IF, Shrink-IF | 0.9728 | 0.8604 | 0.9053 | 0.9136 | 0.9094 | 0.7244 (0.7124,0.7364) |
| Cold-IF, Warm-IF, Shrink-IF | 0.9722 | 0.8611 | 0.9068 | 0.9108 | 0.9088 | 0.7238 (0.7118,0.7358) |
| **AGELFS** | | | | | | |
| Cold-IF, Warm-IF | 0.9731 | **0.8698** | **0.9218** | 0.8920 | 0.9067 | 0.7284 (0.7164,0.7404) |
| Cold-IF, Shrink-IF | **0.9734** | 0.8598 | 0.9026 | **0.9195**$^*$ | **0.9110** | 0.7267 (0.7147,0.7387) |
| Warm-IF, Shrink-IF | 0.9727 | 0.8591 | 0.9020 | 0.9192 | 0.9105 | 0.7254 (0.7134,0.7374) |
| Cold-IF, Warm-IF, Shrink-IF | 0.9729 | 0.8558 | 0.8980 | 0.9229 | 0.9103 | 0.7224 (0.7103,0.7345) |

Bold numerical values denote superior performance in their respective columns. The

$^*$ denotes statistically significant recall ($p < 0.00001$) compared to the baseline.

We select the baseline model based on the best MCC performance reported for the individual models in Table 5. We observe that the Attention-Guided Ensemble with Learnable Fuzzy Softmax (AGELFS) of the Cold-IF and Shrink-IF models deliver significantly superior values for recall ($p < 0.00001$) and marginally higher values for AUPRC and F-score among other ensemble methods. The AGELFS of the Cold-IF and Warm-IF models deliver higher but not significantly superior values for balanced accuracy and precision. The learned Fuzziness values for the Softmax Layer in the AGELFS ensemble are 1.113, 1.113, 1.039, and 1.044 for the model pairs (Cold-IF, Warm-IF), (Cold-IF, Shrink-IF), (Warm-IF, Shrink-IF), and (Cold-IF, Warm-IF, Shrink-IF), respectively. The Equal Weight Averaging (EWA) ensemble of the Warm-IF and Shrink-IF models yields a marginally higher MCC value compared to the baseline.

Table 8 presents the performances achieved with the external adult and pediatric test sets. For brevity, we present here only the key results in a single table while the complete tables are included in the Supplementary (S1 Table, S2 Table, S3 Table, and S4 Table). We observe suboptimal external generalization compared to the results achieved with the internal test set in Table 7. For the external adult test set, the individual Cold-IF model achieves a relatively higher MCC of 0.4378 among other individual models and so we choose it as the baseline. The F-SLSQP ensemble of the Cold-IF and Warm-IF models demonstrates significantly superior precision ($p < 0.00001$) and the highest AUPRC. The EWA ensemble of the Cold-IF and Warm-IF models achieves a marginally higher balanced accuracy, recall, F-score, and MCC compared to the baseline and other tested combinations. For the Ped-2 test set, the Cold-IF model serves as the baseline. The EWA ensemble significantly improves recall ($p < 0.00001$). We use the Warm-IF as the baseline for the Ped-11 test set. The EWA ensemble of Cold-IF, Warm-IF, and Shrink-IF models demonstrates significantly superior values for recall ($p < 0.00001$). The AGELFS of Cold-IF and Warm-IF models demonstrate higher values for precision; however, these values are not markedly different compared to the individual Warm-

**Table 8. Performances achieved with the external adult and pediatric test sets.** Bold numerical values denote superior performance in their respective columns. The * denotes statistical significance for the respective column metric compared to the baseline models for each external test set.

| Models | AUPRC | B. Acc. | P | R | F | MCC |
|---|---|---|---|---|---|---|
| **Adult** | | | | | | |
| Cold-IF-Baseline | 0.8490 | 0.7170 | 0.8452 | 0.5807 | 0.6884 | 0.4378 (0.4226,0.4530) |
| EWA Ensemble | | | | | | |
| Cold-IF, Warm-IF | 0.8557 | **0.7272** | 0.8519 | **0.5976** | **0.7024** | **0.4568 (0.4415,0.4721)** |
| F-SLSQP Ensemble | | | | | | |
| Cold-IF, Warm-IF | **0.8591** | 0.6961 | **0.8929*** | 0.4697 | 0.6156 | 0.4205 (0.4053,0.4357) |
| **Ped-2** | | | | | | |
| Cold-IF-Baseline | **0.4685** | **0.5480** | 0.3997 | 0.8794 | **0.5496** | **0.1206 (0.1118,0.1294)** |
| EWA Ensemble | | | | | | |
| Cold-IF, Warm-IF, Shrink-IF | 0.4367 | 0.5240 | 0.3847 | **0.9335*** | 0.5449 | 0.0785 (0.0713,0.0857) |
| AGELFS | | | | | | |
| Cold-IF, Shrink-IF | 0.4567 | 0.5475 | **0.4003** | 0.8529 | 0.5449 | 0.1135 (0.1050,0.1220) |
| **Ped-11** | | | | | | |
| Warm-IF-Baseline | 0.5381 | **0.6446** | 0.4368 | 0.6976 | 0.5372 | 0.2681 (0.2559,0.2803) |
| EWA Ensemble | | | | | | |
| Cold-IF, Warm-IF | 0.5381 | 0.6222 | 0.3963 | 0.7918 | 0.5282 | 0.2336 (0.2220,0.2452) |
| Cold-IF, Warm-IF, Shrink-IF | 0.5380 | 0.6180 | 0.3924 | **0.7943*** | 0.5253 | 0.2267 (0.2152,0.2382) |
| AGELFS | | | | | | |
| Cold-IF, Warm-IF | 0.5450 | 0.6422 | **0.4374** | 0.6833 | 0.5334 | 0.2636 (0.2515,0.2757) |
| Cold-IF, Shrink-IF | **0.5528** | 0.6420 | 0.4297 | 0.7145 | 0.5367 | 0.2635 (0.2514,0.2756) |
| Cold-IF, Warm-IF, Shrink-IF | 0.5411 | 0.6414 | 0.4236 | 0.7394 | **0.5386** | 0.2631 (0.2510,0.2752) |
| **Ped-18** | | | | | | |
| Warm-IF-Baseline | 0.6820 | 0.7324 | 0.6229 | 0.8241 | 0.7095 | 0.4614 (0.4448,0.4780) |
| EWA Ensemble | | | | | | |
| Cold-IF, Warm-IF, Shrink-IF | 0.6698 | 0.7144 | 0.5890 | **0.8616*** | 0.6997 | 0.4342 (0.4177,0.4507) |
| AGELFS | | | | | | |
| Cold-IF, Warm-IF | 0.6804 | **0.7368** | **0.6237** | 0.8371 | **0.7148** | **0.4708 (0.4542,0.4874)** |
| Cold-IF, Warm-IF, Shrink-IF | **0.6852** | 0.7178 | 0.5939 | 0.8582 | 0.7020 | 0.4397 (0.4232,0.4562) |

IF model, which demonstrates the highest MCC compared to the ensembles. We use the Warm-IF as the baseline for the Ped-18 test set. The EWA ensemble of Cold-IF, Warm-IF, and Shrink-IF models achieves significantly superior recall ($p < 0.00007$), while the AGELFS of Cold-IF and Warm-IF models yield higher, yet not markedly different, balanced accuracy, precision, F-score, and MCC values.

We describe below our assessment of potential reasons for the significant improvement in recall ($p < 0.05$) when using the EWA ensemble of Cold-IF, Warm-IF, and Shrink-IF models to predict the external pediatric test sets:

i. Diverse error patterns: The models in the EWA ensemble exhibit different error patterns for the same classification task. The EWA ensemble excels at identifying true positive (TP) samples and enhancing recall. However, an increase in false positive (FP) predictions could counteract precision improvements, resulting in relatively unchanged F-score, MCC, and AUPRC.

ii. Ensemble learning bias-variance tradeoff: Ensemble learning aims to reduce the bias and variance of individual models for better generalization. The EWA ensemble decreases variance without significantly impacting bias. Since recall is sensitive to reducing false negatives

(FN) (i.e., variance reduction), it can show significant improvement while other metrics remain unchanged if bias remains relatively constant.

iii. Imbalanced datasets: In imbalanced datasets, EWA ensemble techniques can improve recall for the minority class without significantly affecting other metrics. This is evident in the external pediatric test sets where abnormal CXRs are fewer compared to normal samples. The EWA ensemble model's robustness against overfitting and improved generalization in identifying minority class samples may not lead to significant changes in other metrics. The aforementioned discussions also apply to the significantly superior recall values obtained using the AGELFS of Cold-IF and Shrink-IF models for the internal adult test.

## Conclusion and future scope

Diverse model initialization techniques are instrumental for deep model optimization thereby affecting convergence speed, reducing the risk of overfitting, and improving generalizability. Our qualitative and quantitative analyses validate the claim that cold-start approaches can decelerate convergence while warm-start methods, such as ImageNet-pretrained weight initialization, enhance convergence and performance. Furthermore, improper weight initialization can introduce biases that inadvertently favor certain classes or feature sets which, in turn, increases the risk of model overfitting to the data and reducing generalizability. To mitigate this risk, we perform ensemble learning and propose novel weight-level ensemble methods to improve performance over individual constituent models. These ensembles can harness a broader range of feature representations, making them more adaptable and effective when handling unseen data. This adaptability is particularly relevant in medical computer vision, where models must demonstrate exceptional generalizability across diverse patient populations and imaging modalities.

Our study, however, has several limitations: The current study employs the VGG-16 model due to its simplicity and established performance in medical imaging. In future work, we intend to broaden the architectural landscape by incorporating more modern CNN and vision transformer-based architectures. These architectures offer different layers of complexity and specialization that could potentially yield novel insights into the interplay between model initialization and performance. Given the risk of overfitting with improper initialization, incorporating advanced regularization techniques like dropout, L1/L2 regularization, or even Bayesian Neural Networks could offer additional robustness.

Another intriguing direction is to compare the effects of using pre-trained weights from expansive and specialized medical datasets, such as RadImageNet [47], against those initialized with ImageNet. We plan to train models on this dataset from scratch using randomly initialized weights and compare their performance against models initialized with both cold starts and ImageNet pre-trained weights.

While our proposed weight-level ensemble methods demonstrate promising improvements in performance, they also incur a computational overhead that may not be feasible in resource-constrained settings. Future research could focus on optimizing these ensemble techniques for such environments. Utilizing high-end GPUs and cloud-based computing resources can mitigate this limitation to some extent, enabling the deployment of these advanced ensemble techniques in a broader range of settings.

As our models get more complex and effective, understanding their decision-making process becomes crucial. Future work could focus on making these DL models more interpretable to clinicians. While we have employed Bayesian optimization for certain aspects of our model, a more comprehensive AutoML approach could potentially yield even better results.

Extending the scope of initialization techniques to include demographic information like age, sex, and other medical history could pave the way for more personalized DL models in medical imaging. Another exciting avenue is the integration of various medical imaging modalities like MRI, CT scans, and CXRs within a single unified model. This can be particularly useful in complex diagnoses where multiple types of imaging data are often reviewed by physicians. Pursuing these research directions could help improve medical computer vision DL models for reliable healthcare applications.

## Supporting information

**S1 Fig. Confusion matrices for each model pair while predicting the internal adult test set.**
(TIF)

**S2 Fig. MCC comparison along with the *p*-value for each model pair while predicting the internal adult test set.**
(TIF)

**S3 Fig. Histograms show the distribution of Softmax activations for each model pair.** (a) Cold-RF and Cold-IF; (b) Warm-RF and Warm-IF, and (c) Shrink-RF and Shrink-IF.
(TIF)

**S1 Table. Performances achieved with the external adult test.** Bold numerical values denote superior performance in their respective columns. The * denotes statistically significant precision ($p<0.00001$) compared to the baseline.
(DOCX)

**S2 Table. Performances achieved with the external Ped-2 test.** Bold numerical values denote superior performance in their respective columns. The * denotes statistically significant recall ($p<0.00001$) compared to the baseline.
(DOCX)

**S3 Table. Performances achieved with the external Ped-11 test.** Bold numerical values denote superior performance in their respective columns. The * denotes statistically significant recall ($p<0.00001$) compared to the baseline.
(DOCX)

**S4 Table. Performances achieved with the external Ped-18 test.** Bold numerical values denote superior performance in their respective columns. The * denotes statistically significant recall ($p<0.00007$) compared to the baseline.
(DOCX)

## Author Contributions

**Conceptualization:** Sivaramakrishnan Rajaraman, Ghada Zamzmi, Feng Yang, Zhaohui Liang, Zhiyun Xue, Sameer Antani.

**Data curation:** Sivaramakrishnan Rajaraman, Feng Yang.

**Formal analysis:** Sivaramakrishnan Rajaraman, Sameer Antani.

**Funding acquisition:** Sameer Antani.

**Investigation:** Sameer Antani.

**Methodology:** Sivaramakrishnan Rajaraman, Ghada Zamzmi, Feng Yang, Zhaohui Liang, Zhiyun Xue, Sameer Antani.

**Project administration:** Sameer Antani.

**Resources:** Sameer Antani.

**Software:** Sivaramakrishnan Rajaraman, Feng Yang.

**Supervision:** Sameer Antani.

**Visualization:** Sivaramakrishnan Rajaraman.

**Writing – original draft:** Sivaramakrishnan Rajaraman, Sameer Antani.

**Writing – review & editing:** Sivaramakrishnan Rajaraman, Ghada Zamzmi, Feng Yang, Zhao-hui Liang, Zhiyun Xue, Sameer Antani.

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
