## [Decision Letter · Decision Letter 0]

16 Oct 2023

PDIG-D-23-00209

Uncovering the effects of model initialization on deep model generalization: A study with adult and pediatric chest X-ray images

PLOS Digital Health

Dear Dr. Antani,

Thank you for submitting your manuscript to PLOS Digital Health. After careful consideration, we feel that it has merit but does not fully meet PLOS Digital Health's publication criteria as it currently stands. Therefore, we invite you to submit a revised version of the manuscript that addresses the points raised during the review process.

Please submit your revised manuscript within 30 days Nov 15 2023 11:59PM. If you will need more time than this to complete your revisions, please reply to this message or contact the journal office at digitalhealth@plos.org. Please include the following items when submitting your revised manuscript:

We look forward to receiving your revised manuscript.

Kind regards,

Ulas Bagci

Guest Editor

PLOS Digital Health

Journal Requirements:

2. Please ensure that Funding Information and Financial Disclosure Statement are matched.

3. In the Funding Information you indicated that no funding was received. Please revise the Funding Information field to reflect funding received.

4. Please provide separate figure files in .tif or .eps format only and remove any figures embedded in your manuscript file. Please also ensure that all files are under our size limit of 10MB.

Additional Editor Comments (if provided):

the paper gets positive feedbacks from two expert reviewers.

there are some mixed major and minor concerns, but regarding the overall impact and the addressability of the concerns, the manuscript lies in the border of minor to major revision, with more in the minor line.

It would still be great if authors can address all the raised questions.

Reviewers' comments:

Reviewer's Responses to Questions

**Comments to the Author**

1. Does this manuscript meet PLOS Digital Health’s publication criteria? Is the manuscript technically sound, and do the data support the conclusions? The manuscript must describe methodologically and ethically rigorous research with conclusions that are appropriately drawn based on the data presented.

Reviewer #1: Yes

Reviewer #2: Yes

2. Has the statistical analysis been performed appropriately and rigorously?

Reviewer #1: Yes

Reviewer #2: Yes

3. Have the authors made all data underlying the findings in their manuscript fully available (please refer to the Data Availability Statement at the start of the manuscript PDF file)?

Reviewer #1: Yes

Reviewer #2: Yes

4. Is the manuscript presented in an intelligible fashion and written in standard English?

Reviewer #1: Yes

Reviewer #2: Yes

5. Review Comments to the Author

Reviewer #1: The authors' approach to exploring the impact of various weight initialization methods in deep learning models on the classification of chest X-rays is indeed compelling, especially with the focus on the generalizability of a model trained on adult data to the pediatric population. The manuscript presents a good analysis with both empirical and statistical analysis, though a few points would benefit from further clarification:

• The author(s) have Could the authors share the factors that influenced their decision to focus on adult and pediatric CXRs for this study?

• The authors indicate that the training and validation sets from the RSNA dataset were divided into two equal-sized subsets to facilitate the warm-start scenario. Could the authors share any supporting references that guided this split?. 

• The methodology employed for splitting the datasets into training, validation, and testing categories has been mentioned. Could the authors provide additional details on whether the splits were randomly executed or if a specific stratification process was adopted?

• I also would like to know, whether split has been performed as subject-independent or subject dependent? If it’s not the later case, I may suggest the author(s) to justify it. 

• The author(s) have already reported several works related to DNN for adult and pediatric chest x-ray-based diagnosis. How this present work is differ from the models which are developed in the earlier works? Could you justify it?

• It would be good to include some more potential references in the recent year (especially in 2023) and also try to avoid the self-citations, if its not relevant to this present work. 

• It has been noted that pediatric chest X-rays were partitioned into three distinct groups. Could the rationale behind this stratification be elaborated further with some clinical context?

• The decision to resize the images to a resolution of 256x256 for model training is mentioned. Could the authors shed light on the factors that influenced this choice of resolution?

• In eqn 1, the author mentioned a constant value of 0.01 for alpha and beta and it has been referred from the earlier works. How do you justify that the values reported in the earlier works also fit for your present work?

• Why there is no performance comparison between the present work with the relevant works in the literature is not reported?

• Whether the proposed solution in the present work also satisfy the network which is used for multi-class classification using x-ray images? Will it be suitable for other modality based network (ex: CT scan, MRI, etc)

• The choice of the VGG-16 model, albeit with certain modifications, is intriguing. Could the authors clarify the reasoning behind this particular selection and provide insight into how the performance might have been affected if alternate models were employed?

• The utilization of Bayesian optimization for selecting the α value in the ‘shrink and perturb’ model is highlighted. Could the authors elaborate on the factors that informed this decision?

• The choice of F-score-weighted Sequential Least-Squares Quadratic Programming and Attention-Guided Ensembles with Learnable Fuzzy Softmax, among other existing ensemble methods for aggregating weight parameters, is interesting. Could the authors provide more insights into the rationale behind this choice?

• Additional details on the Clopper-Pearson Exact methodology and Z-test performed for the statistical significance analysis would be appreciated. Could the authors specify the null and alternative hypotheses established for these tests?

• The authors mention observing that models initialized with ImageNet-pretrained weights demonstrated superior generalizability over randomly-initialized counterparts, contradicting certain findings for non-medical images. Could the authors share their thoughts on why this difference in results might occur between medical and non-medical images?

• The reported results indicate that weight-level ensembles of these models achieved significantly higher recall. Could the authors elaborate on how this impacts other performance metrics, such as precision or the F1 score?

• Can the authors outline the potential next steps following this research? Are there any other initialization techniques or ensemble methods the authors plan to explore in future studies?

• Include the potential limitations and possible future work in the discussion section.

Reviewer #2: The research compares various initialization methods for convolutional neural network models and examines their impact on convergence rate, performance, and generalizability when applied to chest x-ray images. Despite the distinction between the ImageNet dataset's natural images and medical image features, the models initialized with ImageNet pre-trained weights showcased better performance compared to those with random initialization.

The paper is well-written and easy to follow. The overall experimental setup is well-designed and supports the claims. 

Weaknesses:

- The investigation exclusively employs the VGG framework. Given that VGG is an overparameterized model, it is prone to overfitting sooner than other models. Delving into more recent architectures like ResNest, DenseNet, ConvNext, and others could offer additional insights.

- There is an absence of a comparison involving pre-training weights from expansive radiology datasets, such as RadImageNet. Assessing the relative impacts of ImageNet pre-training versus RadImageNet training could be useful. This aspect might be an avenue for further research.

6. PLOS authors have the option to publish the peer review history of their article (what does this mean?). If published, this will include your full peer review and any attached files.

**Do you want your identity to be public for this peer review?** For information about this choice, including consent withdrawal, please see our Privacy Policy.

Reviewer #1: Yes: Murugappan Murugapan

Reviewer #2: No

---

## [Decision Letter · Decision Letter 1]

4 Dec 2023

Uncovering the effects of model initialization on deep model generalization: A study with adult and pediatric chest X-ray images

PDIG-D-23-00209R1

Dear Dr. Antani,

We are pleased to inform you that your manuscript 'Uncovering the effects of model initialization on deep model generalization: A study with adult and pediatric chest X-ray images' has been provisionally accepted for publication in PLOS Digital Health.

Best regards,

Ulas Bagci

Guest Editor

PLOS Digital Health

Rebuttal period was successful!

Reviewer Comments (if any, and for reference):

Reviewer's Responses to Questions

**Comments to the Author**

1. If the authors have adequately addressed your comments raised in a previous round of review and you feel that this manuscript is now acceptable for publication, you may indicate that here to bypass the “Comments to the Author” section, enter your conflict of interest statement in the “Confidential to Editor” section, and submit your "Accept" recommendation.

Reviewer #1: All comments have been addressed

2. Does this manuscript meet PLOS Digital Health’s publication criteria? Is the manuscript technically sound, and do the data support the conclusions? The manuscript must describe methodologically and ethically rigorous research with conclusions that are appropriately drawn based on the data presented.

Reviewer #1: Yes

3. Has the statistical analysis been performed appropriately and rigorously?

Reviewer #1: Yes

4. Have the authors made all data underlying the findings in their manuscript fully available (please refer to the Data Availability Statement at the start of the manuscript PDF file)?

Reviewer #1: Yes

5. Is the manuscript presented in an intelligible fashion and written in standard English?

Reviewer #1: Yes

6. Review Comments to the Author

Reviewer #1: I appreciate the author(s) efforts in addressing the comments in a more sensible and technical way. This will certainly helps the reader for a better understanding.

7. PLOS authors have the option to publish the peer review history of their article (what does this mean?). If published, this will include your full peer review and any attached files.

**Do you want your identity to be public for this peer review?** For information about this choice, including consent withdrawal, please see our Privacy Policy.

Reviewer #1: No
